# Neo-GNNs: Neighborhood Overlap-aware Graph Neural Networks for Link Prediction

**Seongjun Yun, Seoyoon Kim, Junhyun Lee, Jaewoo Kang**$^*$ **, Hyunwoo J. Kim**$^*$
Department of Computer Science and Engineering
Korea University
{ysj5419, sykim45, ljhyun33, kangj, hyunwoojkim}@korea.ac.kr

## Abstract

Graph Neural Networks (GNNs) have been widely applied to various fields for learning over graph-structured data. They have shown significant improvements over traditional heuristic methods in various tasks such as node classification and graph classification. However, since GNNs heavily rely on smoothed node features rather than graph structure, they often show poor performance than simple heuristic methods in link prediction where the structural information, *e.g.*, overlapped neighborhoods, degrees, and shortest paths, is crucial. To address this limitation, we propose **Ne**ighborhood **O**verlap-aware Graph Neural Networks (Neo-GNNs) that learn useful structural features from an adjacency matrix and estimate overlapped neighborhoods for link prediction. Our Neo-GNNs generalize neighborhood overlap-based heuristic methods and handle overlapped multi-hop neighborhoods. Our extensive experiments on Open Graph Benchmark datasets (OGB) demonstrate that Neo-GNNs consistently achieve state-of-the-art performance in link prediction.

## 1  Introduction

Graph-structured data is ubiquitous in a wide range of domains ranging from social network analysis [1, 2, 3] to biology [4, 5, 6, 7] and computer vision [8, 9, 10]. In recent years, numerous variants of Graph neural networks (GNNs) have been proposed for learning representations over graph-structured data. GNNs learn low dimensional representations of nodes or graphs via iterative aggregation of features from neighbors using non-linear transformations. In this manner, GNNs have shown significant improvements over traditional methods, *e.g.*, heuristic methods and embedding-based methods, and achieved state-of-the-art performance on various tasks, such as node classification [11, 12, 13, 14, 15], graph classification [16, 17, 18, 19, 20], and graph generation [21, 22, 23, 24].

However, in link prediction, traditional heuristic methods still show competitive performance compared to GNNs, which often even outperform GNNs. This is because structural information, (*e.g.*, overlapped neighborhoods, degrees, and shortest path), is crucial for link prediction whereas GNNs heavily rely on smoothed node features rather than graph structure. Recently, SEAL [25] has been proposed to consider structural information for link prediction by utilizing the relative distance between the target node pair and their neighborhoods. Nonetheless, SEAL requires the expensive computational cost to apply a GNN independently to an extracted subgraph for each target node pair.

To address this limitation, we propose **Ne**ighborhood **O**verlap-aware Graph Neural Networks (Neo-GNNs) that are designed to consider key structural information regarding links without manual processes. Specifically, instead of using input node features, Neo-GNNs first learn to generate useful structrual features for each node from an adjacency matrix. Then Neo-GNNs measure the existence of links by considering the structural features of overlapped neighborhoods via neighborhood

---

$^*$corresponding author

35th Conference on Neural Information Processing Systems (NeurIPS 2021).

overlap-aware aggregation scheme. Finally, to consider both structural information and input node features, our proposed model adaptively combines scores from Neo-GNNs and feature-based GNNs in an end-to-end fashion. We show that Neo-GNNs consistently outperform both state-of-the-art GNNs and heuristic methods on four Open Graph Benchmark datasets (OGB) for link prediction. Furthermore, Our Neo-GNNs generalize the neighborhood overlap-based heuristic methods which measure the likelihood of the link based on manually designed structural information of overlapped neighbors.

Our **contributions** are as follows: (i) We propose **Ne**ighborhood **O**verlap-aware Graph Neural Networks (Neo-GNNs) that learn useful structural features from an adjacency matrix and estimate overlapped neighborhoods for link prediction. (ii) Neo-GNNs generalize neighborhood overlap-based heuristic methods and handle overlapped multi-hop neighborhoods. (iii) Our extensive experiments on Open Graph Benchmark datasets (OGB) demonstrate that Neo-GNNs consistently achieve state-of-the-art performance in link prediction.

## 2   Related Works

**Graph Neural Networks.** GNNs have been designed to learn node representations by using neural networks on graph topology. Among deep learning based approaches, the message passing scheme is dominantly used in recent studies such as GCN [11], GraphSAGE [26], and GAT [12]. Due to the iterative aggregation step, each node representation vector can have information of neighbor nodes in multi-hop relationships required for downstream tasks. However, there is a limitation of the expressive power that is upper-bounded by the 1-Weisfeiler-Lehman (1-WL) graph isomorphism test. To overcome this limitation, recent works have tried to boost the expressive power of GNNs by augmenting node features with ordering vectors or position-aware vectors [11, 27, 28]. The main purpose of these works is to complement GNNs with structural information which is crucial for prediction tasks. Our study focuses on adaptively incorporating structural information to GNNs for the link prediction task.

**Link Prediction.** Link prediction has been studied in various ways. Conventionally, diverse heuristic methods have been proposed for link prediction. They basically measure the scores of given node pairs based on structural information *e.g.,* overlapped neighbors and shortest path, about the pair of nodes. Common neighbors and preferential attachment [29] exploit structural information about one-hop neighbors to compute the score. To consider more than one-hop relationships, second-order heuristic methods (*e.g.,* Adamic-Adar [30] and resource allocation [31]) and higher-order heuristic methods (*e.g.,* Katz [32] , PageRank [33] , and SimRank [34]) have been proposed. Heuristic methods are extremely effective for link prediction. However, they require manually designed structural information for each heuristic method. To overcome this limitation, embedding-based methods have been proposed. They learn node embeddings based on connections between nodes and compute similarity scores using the embeddings. Typically, Matrix factorization [35] learns node embeddings by decomposing an adjacency matrix of the graph. Random walk-based embedding methods such as Deepwalk [36], and node2vec [37] learn node embeddings by applying the Skip-Gram [38] techniques on the random walks. LINK [39] learns to classify the existence of links based on each row in the adjacency matrix, which includes connectivity information. Since the performance of the embedding methods depends on the sparsity of the input graph, it is hard to regard these methods as generalized ones. Recently, with the success of GNNs in learning graph representations, there have been several attempts to apply them to the link prediction task. Typically, GAE and VGAE [40] learn node representations through GCN to reconstruct the input graph in the auto-encoder framework. Based on the GAE, various GNN architectures have been applied to link prediction. On the other hand, SEAL [25] reformulated the link prediction task to the classification of enclosing subgraphs. Instead of directly predicting the link, enclosing graphs are sampled around each target link to compose dataset and SEAL performs the graph classification task. Due to the node labeling step to mark nodes' different roles in an enclosing subgraph, SEAL has better performance than GAE even though both are GNN-based methods. However, constructing subgraphs is inefficient because it requires a large amount of computation, whereas our model is as efficient as GAE and can consider structural information like SEAL.

# 3 Methods

The goal of our framework, **Ne**ighborhood **O**verlap-aware Graph Neural Networks (Neo-GNNs), is to learn useful structural features from an adjacency matrix and estimate overlapped neighbors for link prediction. We begin with defining the basic notions of graph neural networks for link prediction and review neighborhood overlap-based heuristic methods, and then introduce Neo-GNNs.

## 3.1 Preliminaries

**Notations.** Consider an undirected graph $\mathcal{G} = (\mathcal{V}, \mathcal{E})$ with $N$ nodes, where $\mathcal{V} = \{v_1, v_2, \ldots, v_N\}$ represents a set of nodes and $\mathcal{E} = \{e_{ij} \mid v_i, v_j \in \mathcal{V}\}$ represents a set of edges where the nodes $v_i, v_j \in \mathcal{V}$ are connected. The adjacency matrix $A \in \mathbf{R}^{N \times N}$ is defined by $A_{ij} = 1$ if $e_{ij} \in \mathcal{E}$ and 0 otherwise. The degree matrix $D \in \mathbf{R}^{N \times N}$ is a diagonal matrix defined by $D_{ii} = \sum_j A_{ij}$. The nodes of $\mathcal{G}$ have their own feature vectors $x_i \in \mathbf{R}^F$ ($i \in \{1, 2, \ldots, N\}$), with $X \in \mathbf{R}^{N \times F}$ denoting the collection of such vectors in a matrix form.

**Graph Neural Networks for Link Prediction.** Given a graph $\mathcal{G}$ and a feature matrix $X$, graph neural networks learn meaningful node representations by an iterative aggregation of transformed representations of neighbor nodes in each $l$-th GNN layer as follows:

$$H^{(l+1)} = \sigma \left( \tilde{A}_{\mathrm{GNN}} H^{(l)} W^{(l)} \right), \tag{1}$$

where $\tilde{A}_{\mathrm{GNN}} \in \mathbf{R}^{N \times N}$ is the adjacency matrix normalized in different ways depending on each GNN architecture (*e.g.,* $\tilde{D}^{-\frac{1}{2}}(A + I)\tilde{D}^{-\frac{1}{2}}$), $W^{(l)} \in \mathbf{R}^{d^{(l)} \times d^{(l+1)}}$ is a trainable weight matrix, and $H^{(0)}$ is the node feature matrix $X \in \mathbf{R}^{N \times F}$. After stacking $L$ GNN layers, node representations $H^{(L)}$ are then used to predict existence of each link $(i, j)$:

$$\hat{y}_{ij} = \sigma(s(h_i^{(L)}, h_j^{(L)})), \tag{2}$$

where $s(\cdot, \cdot)$ is a function, *e.g.,* inner product or MLP, and $h_i^{(L)}$ is the representation of the node $i$ from $H^{(L)}$.

## 3.2 Neighborhood Overlap-based Heuristic Methods

Heuristic methods for link prediction measure the score of given node pairs based on structural information about the node pairs, *e.g.,* shortest path, degree, and common neighbors. Although GNNs outperform existing traditional heuristic methods in various graph tasks, in link prediction, since GNNs heavily rely on smoothed node features rather than graph structure, the heuristic methods often show competitive performance compared to GNNs. Especially, neighborhood overlap-based heuristic methods are straightforward yet highly effective, even better than GNN models in several datasets, *e.g.,* ogbl-collab and ogbl-ppa. Typical neighborhood overlap-based heuristic methods are Common Neighbors, Resource Allocation (RA) [31], and Adamic Adar [30]. The Common Neighbors method measures the score of link $(u, v)$ by counting the number of common neighbors between node $u$ and $v$ as

$$S_{CN}(u, v) = |\mathcal{N}(u) \cap \mathcal{N}(v)| = \sum_{k \in \mathcal{N}(u) \cap \mathcal{N}(v)} 1. \tag{3}$$

The Common Neighbors method is simple and effective, but has a limitation that equally weighs the importance of each common neighbor. To solve this issue, several heuristic methods *e.g.,* Resource Allocation, and Adamic-Adar measure the score for the link by considering the importance of each common neighbor. From the intuition that neighbor nodes with lower degrees are more significant, they give more weight to neighbors with lower degrees. Specifically, Resource Allocation (RA) [31] measures the score of link $(u, v)$ by counting the inverse degrees of common neighbors between node $u$ and $v$ as

$$S_{RA}(u, v) = \sum_{k \in \mathcal{N}(u) \cap \mathcal{N}(v)} \frac{1}{d_k}, \tag{4}$$

where $d_k$ denotes the degree of node $k$. Adamic-Adar has a relatively decreased penalty for higher degree compared to RA by using the reciprocal logarithm of common neighbors' degrees between

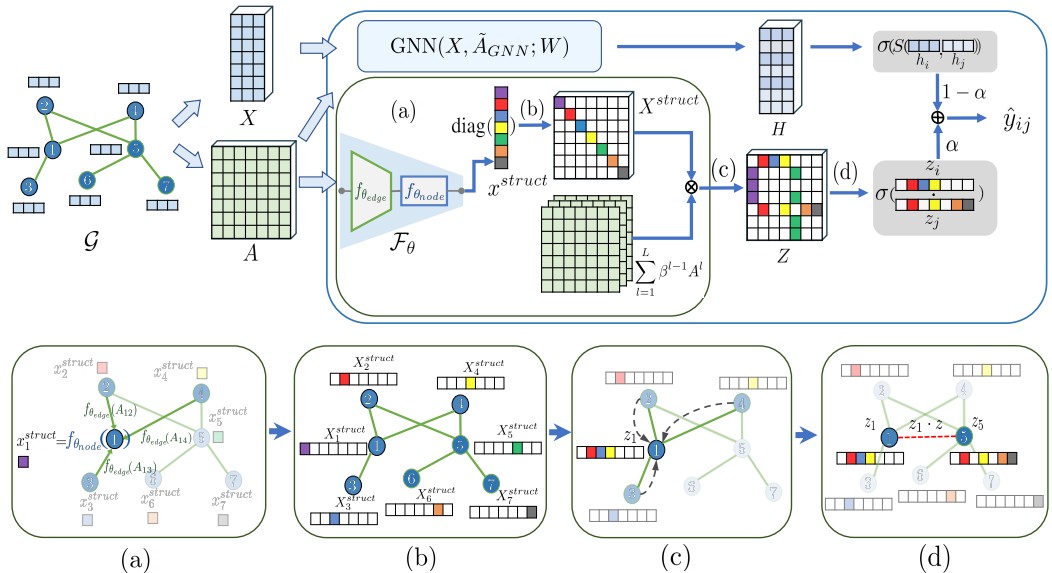

(a)        (b)        (c)        (d)

Figure 1: The Neo-GNNs framework for link prediction. Neo-GNNs learn useful structural features from an adjacency matrix and estimate similarity scores based on overlapped neighborhoods. (a) Neo-GNNs first generate the structural feature vector $x^{struct} \in \mathbf{R}^{N \times 1}$ from an adjacency matrix $A \in \mathbf{R}^{N \times N}$ by using Structural feature generator $\mathcal{F}_\theta$, i.e., $\mathcal{F}_\theta(A)$. Then to consider only features of overlapped neighbors between nodes, (b) Neo-GNNs construct a diagonal matrix $X^{struct} \in \mathbf{R}^{N \times N}$ and (c) aggregate the features of multi-hop neighborhoods by multiplying the sum of powers of adjacency matrices, i.e., $\sum_{l=1}^{L} \beta^{l-1} A^l$. Finally, two node representations $Z$ and $H$, respectively from Neo-GNNs and feature-based GNNs, are used to (d) compute similarity scores and combined adaptively with the learnable parameter $\alpha$.

node $u$ and $v$ as

$$S_{AA}(u, v) = \sum_{k \in \mathcal{N}(u) \cap \mathcal{N}(v)} \frac{1}{\log d_k}. \tag{5}$$

These heuristic methods show comparable performance to GNNs for link prediction.

However, they have two limitations. First, each heuristic method uses manually designed structural features of neighborhoods, *e.g.*, $1, \frac{1}{d}, \frac{1}{\log d}$. This requires the manual choice by domain experts to select the best heuristic method for each dataset. Second, they only consider structural similarity. While the GNNs do not use graph structures well compared to using node features, heuristic methods cannot utilize the node features for link prediction.

To address the limitations of both GNNs and heuristic methods, we propose Neighborhood Overlap-aware Graph Neural Networks (Neo-GNN), that learn useful structural features from an adjacency matrix and estimate overlapped neighborhoods for link prediction, and adaptively combine with the conventional feature-based GNNs in an end-to-end fashion.

### 3.3 Neighborhood Overlap-aware Graph Neural Networks

We now introduce Neighborhood Overlap-aware Graph Neural Networks (Neo-GNNs) for link prediction. We first explain how Neo-GNNs learn and utilize structural information for link prediction and then explain the process of adaptively combining with the feature-based GNNs.

Neo-GNNs consist of two key components: (1) Structural feature generator and (2) neighborhood overlap-aware aggregation scheme. First, as we discussed in section 3.2, each heuristic method uses manually designed structural features of neighborhoods, $1, \frac{1}{d}, \frac{1}{\log d}$. To generalize and learn these structural features, we propose Structural feature generator $\mathcal{F}_\theta$ which learns to generate structural

features of each node using an only adjacency matrix $A \in \mathbf{R}^{N \times N}$ of the graph as

$$x_i^{struct} = \mathcal{F}_\theta(A_i) = f_{\theta_{node}} \left( \sum_{j \in \mathcal{N}_i} f_{\theta_{edge}}(A_{ij}) \right), \tag{6}$$

where $x_i^{struct}$ is a structural feature value of the node $i$ and $\mathcal{F}_\theta$ is a learnable function comprised of two MLPs, $f_{\theta_{node}}$ and $f_{\theta_{edge}}$, for nodes and edges, respectively. That is to say, Neo-GNNs take only an adjacency matrix $A$ as an input to generate the most beneficial structural features. This input adjacency matrix $A$ can be replaced with the combination of powers of adjacency matrices. Now, Structural feature generator $\mathcal{F}_\theta$ can generate structural features for each heuristic method. For example, if $f_{\theta_{node}}$ is a reciprocal of the logarithm function, i.e., $f(x) = \frac{1}{\log x}$, and $f_{\theta_{edge}}$ is an identity function, i.e., $f(x) = x$, then Structural feature generator $\mathcal{F}_\theta$ can generate the exactly same structural feature as the features used in Adamic-Adar method.

Based on the generated structural features of each node, the next process is to calculate the similarity score that considers only structural features of overlapped neighbors between given nodes. Note that conventional GNNs cannot compute this score due to two reasons: the normalized adjacency matrix and the lower dimension of hidden representations than the number of nodes (i.e., $d \ll N$). The normalized adjacency matrix hinders GNNs counting a number of neighborhoods and the low dimension makes features of each neighborhoods indistinguishable after aggregation, which cannot detect the neighborhoods overlap. We propose the neighborhood overlap-aware aggregation scheme to calculate neighborhood overlap-aware score. First, to maintain the respective features of each node after aggregation, we construct a diagonal matrix $X^{struct} \in \mathbf{R}^{N \times N}$ using the structural feature vector $x^{struct} \in \mathbf{R}^{N \times 1}$ as

$$X^{struct} = \text{diag}(x^{struct}). \tag{7}$$

Then, to consider the number of overlapped neighbors, we aggregate features of neighborhoods by multiplying an unnormalized adjacency matrix $A$ as

$$Z = AX^{struct}. \tag{8}$$

Now, each $i$-th row vector of $Z$, $z_i$, involves all the features of node $i$'s neighboring nodes individually. If we compute the inner product of two row vectors in $Z$, then we can compute the scores with the only overlapped neighborhoods, which equals the sum of square of structural feature values of overlapped neighborhoods, i.e., $z_i^T z_j = \sum_{k \in \mathcal{N}(i) \cap \mathcal{N}(j)} \left(x_k^{struct}\right)^2$.

Furthermore, to consider multi-hop overlapped neigbhors, we extend (8) to multi-hop settings as follows:

$$Z = g_\Phi \left( \sum_{l=1}^{L} \beta^{l-1} A^l X^{struct} \right), \tag{9}$$

where $\beta$ denotes a hyper-parameter controlling how much weight is given to close neighbors versus distant neighbors and $g_\Phi$ is a MLP which controls the scale of representations $Z$. Since node representations $Z$ are based on only structural information, we compute feature-based node representations $H \in \mathbf{R}^{N \times d'}$ using the conventional feature-based GNNs as

$$H = \text{GNN}(X, \tilde{A}_{GNN}; W), \tag{10}$$

where $X \in \mathbf{R}^{N \times F}$ denotes the raw feature matrix, $\tilde{A}_{GNN}$ denotes a normalized adjacency matrix, and $W$ is the parameter for GNNs. Then given a link $(i, j)$, Neo-GNNs calculate both similarity scores from each representation matrix $Z$ and $H$ and compute the convex combination of two scores by a trainable parameter $\alpha$ as follows:

$$\hat{y}_{ij} = \alpha \cdot \sigma(z_i^T z_j) + (1 - \alpha) \cdot \sigma(s(h_i, h_j)), \tag{11}$$

Based on (12), we jointly train our proposed model and individual models using three standard (binary) cross-entropy losses

$$\mathcal{L} = \sum_{(i,j) \in D} \left( \lambda_1 BCE(\hat{y}_{ij}, y_{ij}) + \lambda_2 BCE(\sigma(z_i^T z_j), y_{ij}) + \lambda_3 BCE(\sigma(s(h_i, h_j)), y_{ij}) \right), \tag{12}$$

where $BCE(\cdot, \cdot)$ denotes binary cross entropy loss and $\lambda_i$ are the weights.

Table 1: Statistics and evaluation metrics of OGB link prediction datasets.

| Dataset | #Nodes | #Edges | Avg. node deg. | Density | Split ratio | Metric |
|---------|--------|--------|----------------|---------|-------------|--------|
| OGB-PPA | 576,289 | 30,326,273 | 73.7 | 0.018% | 70/20/10 | Hits@100 |
| OGB-COLLAB | 235,868 | 1,285,465 | 8.2 | 0.0046% | 92/4/4 | Hits@50 |
| OGB-DDI | 4,267 | 1,334,889 | 500.5 | 14.67% | 80/10/10 | Hits@20 |
| OGB-CITATION2 | 2,927,963 | 30,561,187 | 20.7 | 0.00036% | 98/1/1 | MRR |

**Computational Complexity.**  Our proposed model uses the $N \times N$ matrix $X^{struct}$ for neigborhood overlap detection, which generally takes $O(N|\mathcal{E}|_L)$ computational time to compute node representations $Z$ in (9), where $|\mathcal{E}|_L$ denotes a number of edges connected up to $L$-hop. To solve this high complexity issue, we represent matrices $X^{struct}$ and $Z$ as the sparse matrix form and the computational time becomes just $O(|\mathcal{E}|_L)$. Also, as we can pre-compute the set of adjacency matrices $\{A^l\}_{l=1}^L$ in (9), thus there is no additional cost to calculate the powers of the adjacency matrix during training and inference.

# 4  Experiments

In this section, we evaluate the benefits of our method against state-of-the-art models on link prediction benchmarks. Then we analyze the contribution of each component in Neo-GNNs and show how Neo-GNNs can actually generalize and learn neighborhood overlap-based heuristic methods.

## 4.1  Experiment Settings

**Datasets.** We evaluate the effectiveness of our Neo-GNNs for link prediction on Open Graph Benchmark datasets [41] (OGB) : OGB-PPA, OGB-Collab, OGB-DDI, OGB-Citation2. Note that OGB-Collab contains multiple edges. Detailed statistics of each dataset are summarized in Table 1.

**Evaluation.** The evaluation for link prediction is based on the ranking performance of positive test edges over negative test edges. Specifically, in OGB-PPA, OGB-Collab, OGB-DDI, each model ranks positive test edges against randomly-sampled negative edges, and computes the ratio of positive test edges that are ranked at K-th place or above (Hits@K). In OGB-Citation2, the evaluation metric is Mean Reciprocal Rank (MRR), where the reciprocal rank of the true link among the negative candidates is calculated for each source node, and then the average is taken over all source nodes.

**Baselines.** To demonstrate the effectiveness of our Neo-GNNs in link prediction, we compare Neo-GNNs with three heuristic link prediction methods, three embedding-based methods, and five GNN-based models. For heuristic methods, we used three well-known neighborhood-overlap based heuristic methods, Common Neighbors, Adamic Adar [30], and Resource Allocation [31]. Without learning process, they predict links by utilizing each designed structural information regarding overlapped neighborhoods. For embedding-based methods, we used Matrix Factorization, Node2Vec [42], and Multi-Layer Perceptron (MLP). Furthermore, we compare our method to GNN-based models, GCN [11], GraphSAGE [26], JK-Net [43], GAT [12], and SEAL [25]. GCN, GraphSAGE, JK-Net, and GAT compute representations for each node and predict target links by measuring the similarity score between the source and target node of the target links. SEAL extracts enclosing subgraphs around target links and predict target links based on representations of the enclosing subgraphs as graph classification.

**Implementation Details.** We reimplemented neighborhood overlap-based heuristic method,i.e., Common neighbors, Adamic Adar, and Resource allocation from the referenced papers by using PyTorch. For Node2Vec, GCN, GraphSAGE, JK-Net, and GAT, we used the implementation in PyTorch Geometric [44] and the implementation in the official github repository for SEAL. We set the number of layers to 3 and latent dimensionality to 256 for all GNN-based models. To train our method, we used GCN as a feature-based GNN based model and all MLP models in our Neo-GNNs consist of 2 fully connected layers. We jointly trained feature-based GNNs and Neo-GNNs. Since a GNN model requires more epochs for convergence than that of Neo-GNNs on OGB-PPA, and OGB-DDI, we adopted pre-trained GCN to handle this issue. In OGB-Citation2, due to memory issue, we fix the $f_{\theta_{edge}}$ as the identity function. For fair comparison, we reported performances of

Table 2: Link prediction performances (%) of our Neo-GNNs and baselines on Open Graph Benchmark (OGB) datasets. Each number is the average performance for 10 random initialization of the experiments. OOM denotes 'out of memory'. **Bold** indicates the second best performance and underline indicates the best performance.

| Method | OGB-PPA | OGB-COLLAB | OGB-DDI | OGB-CITATION2 |
|---|---|---|---|---|
| Common Neighbors | $27.65 \pm 0.00$ | $50.06 \pm 0.00$ | $17.73 \pm 0.00$ | $76.20 \pm 0.00$ |
| Adamic Adar | $32.45 \pm 0.00$ | $53.00 \pm 0.00$ | $18.61 \pm 0.00$ | $76.12 \pm 0.00$ |
| Resource Allocation | $\underline{\mathbf{49.33}} \pm 0.00$ | $52.89 \pm 0.00$ | $6.23 \pm 0.00$ | $76.20 \pm 0.00$ |
| Matrix Factorization | $27.83 \pm 2.02$ | $38.74 \pm 0.30$ | $17.92 \pm 3.57$ | $53.08 \pm 4.19$ |
| Node2Vec | $17.24 \pm 0.76$ | $41.36 \pm 0.69$ | $21.95 \pm 1.58$ | $53.47 \pm 0.12$ |
| MLP | $0.47 \pm 0.05$ | $19.98 \pm 0.96$ | N/A | $28.99 \pm 0.16$ |
| GCN | $16.98 \pm 1.33$ | $47.01 \pm 0.79$ | $44.60 \pm 8.87$ | $84.79 \pm 0.24$ |
| GraphSAGE | $13.93 \pm 2.38$ | $48.60 \pm 0.46$ | $48.01 \pm 9.02$ | $82.64 \pm 0.01$ |
| JK-Net | $11.40 \pm 2.04$ | $48.84 \pm 0.83$ | $\mathbf{57.98} \pm 6.88$ | OOM |
| GAT | OOM | $44.89 \pm 1.23$ | $29.51 \pm 6.40$ | OOM |
| SEAL | $48.15 \pm 4.17$ | $\mathbf{54.37} \pm 0.02$ | $26.25 \pm 6.00$ | $\mathbf{86.32} \pm 0.52$ |
| Neo-GNN | $\mathbf{49.13} \pm 0.60$ | $\underline{\mathbf{57.52}} \pm 0.37$ | $\underline{\mathbf{63.57}} \pm 3.52$ | $\underline{\mathbf{87.26}} \pm 0.84$ |

all baselines and our Neo-GNNs as the mean and the standard deviation of performances from 10 independent runs, where each seed is from 0 to 9. The experiments are conducted on a RTX 3090 (24GB) and a Quadro RTX (48GB).

## 4.2 Results on Link Prediction

Table 2 shows link prediction results of the baselines and Neo-GNNs on Open Graph Benchmark (OGB) datasets. We use GCN to adaptively combine with Neo-GNNs across all datasets except OGB-Citation2. In OGB-Citation2, since GCN requires 46 GB memory to train, we trained our Neo-GNNs without GCN. As shown in Table 2, we can observe that Neo-GNNs consistently achieve state-of-the-arts performance across all datasets. Especially, Neo-GNNs show significant improvements on OGB-Collab and OGB-DDI, where the improvements of Neo-GNNs over the best baseline are 5.4% and 9.6%, respectively. Furthermore, note that Neo-GNNs achieved state-of-the-art performance in OGB-Citation2 without GCN, that is, by using only graph structures without input node features. Interestingly, conventional feature-based GNNs show poor performance with a huge gap than that of neighborhood overlap-based heuristic methods on OGB-PPA and OGB-Collab. This implies that feature-based GNNs have a difficulty in directly utilizing structural information e.g., degree and overlapped neighbors, for link prediction. According to this implication, Neo-GNNs and SEAL are able to learn structural information, thus these methods accomplish better performance than conventional GNNs do. Moreover, Neo-GNNs and SEAL even show good performance compared to the heuristic methods in all datasets as they can capture structural information that the heuristic methods utilize. Although SEAL shows good performance compared to heuristic methods, SEAL shows poor performance than feature-based GNNs in OGB-DDI. One possible interpretation is that SEAL cannot adaptively utilize the input node features and structural features according to each data. Instead, Neo-GNNs adaptively combine Neo-GNNs and GCN for each dataset using the learnable parameter $\alpha$, which shows even higher performance than each performance of Neo-GNNs and GCN. We further analyze the effectiveness of $\alpha$ in 4.2

## 4.3 Ablation Studies

We present ablation experiments to identify the benefits of different components of Neo-GNNs. First, we evaluate our Neo-GNNs without GCN and examine the effectiveness of the parameter $\alpha$ that adaptively combine scores from Neo-GNNs and GCN. Then we study the effects of considering multi-hop overlapped neighborhoods. Specifically, we investigate the effectiveness of two hyper-parameters, the decaying factor $\beta$ and the maximum hop $L$, related to multi-hop overlapped neighborhoods.

Table 3: Ablation study analyzing the significance of Neo-GNNs on the OGB-PPA, OGB-Collab, OGB-DDI, and OGB-Citation2 datasets for link prediction. $\alpha$ denotes the attention weight of Neo-GNNs' scores.

| Dataset | $\alpha$ | Neo-GNN (w/ GCN) | Neo-GNN (w/o GCN) | GCN |
|---------|----------|------------------|-------------------|-----|
| PPA | $0.98 \pm 0.003$ | $49.13 \pm 0.60$ | $48.63 \pm 0.88$ | $16.98 \pm 1.33$ |
| COLLAB | $0.57 \pm 0.130$ | $57.52 \pm 0.37$ | $55.70 \pm 0.24$ | $47.01 \pm 0.79$ |
| DDI | $0.48 \pm 0.015$ | $63.57 \pm 3.52$ | $17.38 \pm 4.05$ | $44.60 \pm 8.87$ |
| CITATION2 | N/A | OOM | $87.26 \pm 0.84$ | $84.79 \pm 0.24$ |

Figure 2: Link prediction results on the OGB-Collab dataset by varying the maximum hop $L$ (left) and the decaying factor $\beta$ (right).

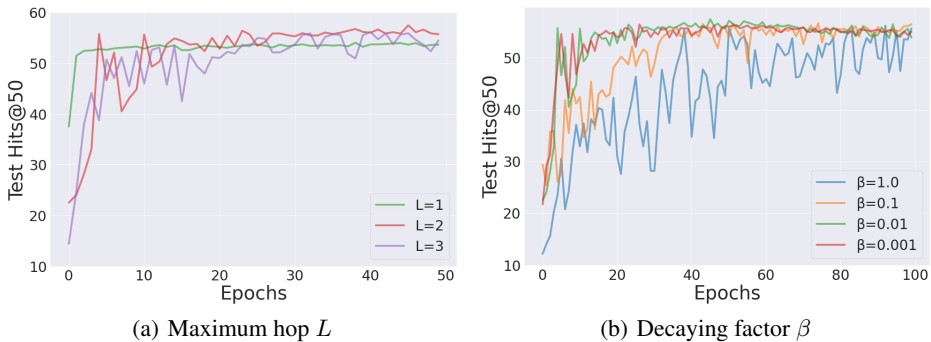

(a) Maximum hop $L$          (b) Decaying factor $\beta$

**Neo-GNNs without GCN.** To measure the effectiveness of Neo-GNNs itself, we perform an ablation study on four datasets, as shown in Table 3. We can see that Neo-GNNs (w/o GCN) still show the state-of-the-art performances compared to baselines except OGB-DDI. Note that Neo-GNNs (w/o GCN) only use graph structures and outperform other GNNs whereas other GNNs use both input features and graph structures. This shows that utilizing key structrual information about overlapped neighbors is crucial for link prediction.

**Effectiveness of the parameter** $\alpha$**.** To consider both structural information and input features, our proposed model predict similarity scores from the convex combination of two scores from Neo-GNNs and GCN by the trainable parameter $\alpha$. As shown in Table 3, $\alpha$ varies for each dataset, which indicates that $\alpha$ properly adjusts the weight of structural information and features for each dataset. With the combining process, Neo-GNNs (w/ GCN) consistently show better performance than performances of individual model, i.e., Neo-GNN (w/o GCN) and GCN. Especially, in OGB-DDI, Neo-GNN (w/o GCN) and GCN show less than 50, but the combined model Neo-GNN (w/ GCN) shows 42% and 80% improved performance compared to each model.

**Effectiveness of multi-hop overlapped neighborhoods.** We study the effectiveness of multi-hop overlapped neighborhoods by investigating effects of two hyper-parameters, $L$ and $\beta$, on OGB-Collab dataset. First, as shown in Figure 2, if Neo-GNNs only consider 1-hop ovelapped neighbhors, i.e., $L = 1$, Neo-GNNs converge faster than other cases considering multi-hop overlapped neighbors. However, the best performance is lower than the others, which shows that multi-hop overlapped neighborhoods enhance the performance of Neo-GNNs for link prediction. Second, $\beta$ controls how much to reduce the effects of neighborhoods when the distance increases. As shown in Figure 2, as $\beta$ decreases, Neo-GNNs converge slowly but eventually show similar performance. This means that if multi-hop overlapped neighbors are informative, then Neo-GNNs achieve good performance robustly to the value of $\beta$.

### 4.4 Analysis on learning neighborhood overlap-based heuristic methods

As we discussed in Sec 3.3, our Neo-GNNs generalize several neighborhood overlap-based heuristic methods, (e.g., Common Neighbors, Adamic Adar, and Resource Allocation). Further, in this section,

we show that Neo-GNNs (w/o GCN) directly learn each neighbhorhood overlap-based heuristic method and implicitly learn the best one among three heuristic methods on OGB-PPA dataset. We first trained Neo-GNNs to fit the scores from each heuristic method on the train set of OGB-PPA. Then we measure the Spearman correlations by using ranks of test edges from each model. We analyze rank correlations between Neo-GNNs and three heuristic methods based on 50000 sampled test edges in Figure 3. As shown in Figure 3, Neo-GNNs show a strong correlation with other heuristic methods. That is, Neo-GNNs can learn each heuristic method. Next, to show that Neo-GNNs learn the most desirable heuristic method depending on datasets, we compute Spearman correlations between ranks from trained Neo-GNNs on OGB-PPA dataset and the heuristic methods. As a result, correlation scores between Neo-GNNs and Resource Allocation, Adamic Adar, and Common Neighbors are 0.9627, 0.9277, and 0.8982, respectively. We can see that correlation scores are proportional to performances of each heuristic method (49.33, 32.45, and 27.65), which learn the best heuristic method.

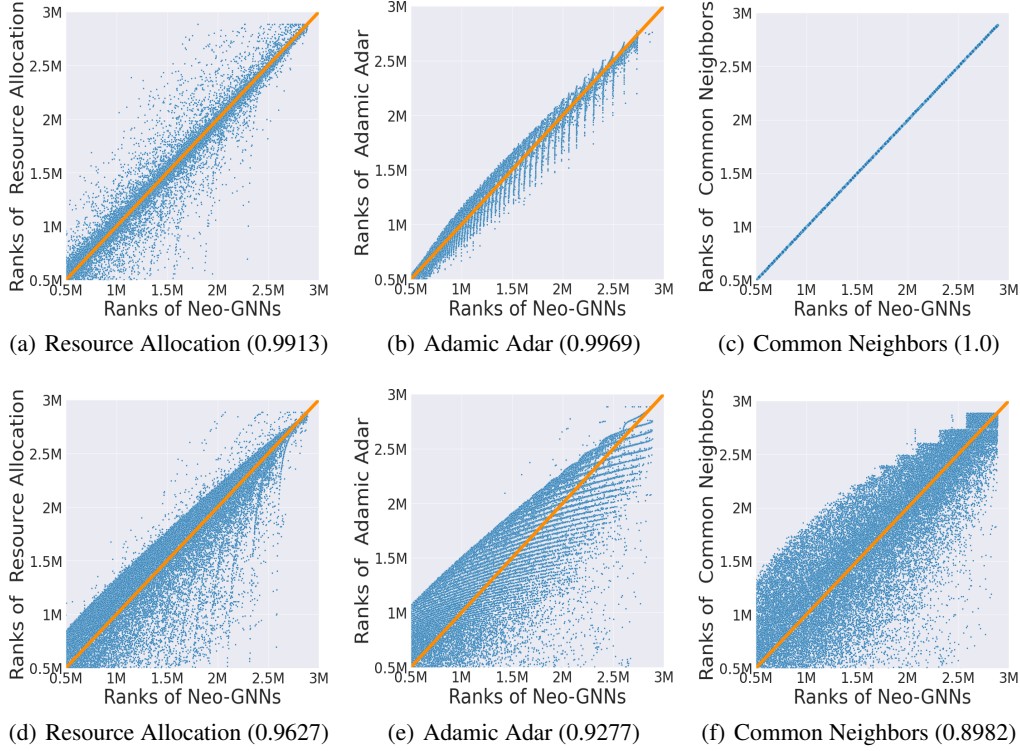

Figure 3: Comparison of rank correlation between our Neo-GNNs and three neighborhood overlap-based heuristic methods on OGB-PPA. We evaluate a rank correlation on positive test edges after ranking the entire test edge prediction scores. We visualize a rank correlation using randomly sampled 50,000 positive test edges. The 3(a), 3(b), and 3(c) show rank correlation when Neo-GNNs (w/o GCN) fit to scores of each heuristic method. The 3(d), 3(e), and 3(f) present the rank correlation between Neo-GNNs (w/o GCN) and each heuristic method upon OGB-PPA. The number in parentheses indicates Spearman Correlation coefficient.

## 5    Conclusion

We introduced Neighborhood Overlap-based Graph Neural Networks (Neo-GNNs) that learn and utilize structural information, which is a key element in link prediction. Neo-GNNs learn useful structural features from an adjacency matrix and estimate overlapped neighborhoods for link prediction. We also adaptively combine Neo-GNNs and feature-based GNNs to consider both structural features and input node features. Furthermore, our Neo-GNNs generalize several neigbhorhood overlap-based heuristic methods and handle overlapped multi-hop neigbhorhoods. Extensive experiments on four Open Graph Benchmark (OGB) datasets demonstrate that Neo-GNNs consistently acheive state-

of-the-art performance on four OGB datasets in link prediction. In future work, we plan to further develop Neo-GNNs to generalize more link prediction-based heuristic methods and improve the scalability with efficient sparse matrix computation.

## 6 Acknowledgement

This work was supported by the following funding sources: National Research Foundation of Korea (NRF-2020R1A2C3010638, NRF-2014M3C9A3063541); ICT Creative Consilience program(IITP-2021-2020-0-01819) supervised by the IITP; Samsung Research Funding & Incubation Center of Samsung Electronics under Project Number SRFC-IT1701-51.

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
