Table 1: Link prediction performances (%) of our Neo-GNNs and heuristic methods on traditional link prediction datasets. Each number is the average performance for 10 random initialization of the experiments. **Bold** indicates the second best performance and **underline** indicates the best performance.

| Data | CN | Jac. | AA | RA | PA | Katz | PR | SR | Neo-GNN (w/o GCN) | Neo-GNN |
|---|---|---|---|---|---|---|---|---|---|---|
| USAir | $93.80 \pm 1.22$ | $89.79 \pm 1.61$ | $95.06 \pm 1.03$ | $95.77 \pm 0.92$ | $88.84 \pm 1.45$ | $92.88 \pm 1.42$ | $94.67 \pm 1.08$ | $78.89 \pm 2.31$ | **96.10** $\pm 0.79$ | **95.56** $\pm 0.57$ |
| Power | $58.80 \pm 0.88$ | $58.79 \pm 0.88$ | $58.79 \pm 0.88$ | $58.79 \pm 0.88$ | $44.33 \pm 1.02$ | $65.39 \pm 1.55$ | $66.00 \pm 1.59$ | **76.15** $\pm 1.06$ | **76.17** $\pm 0.87$ | $72.42 \pm 1.34$ |
| Router | $56.43 \pm 0.52$ | $56.40 \pm 0.00$ | $56.43 \pm 0.51$ | $56.43 \pm 0.51$ | $47.58 \pm 1.47$ | $38.62 \pm 1.35$ | $38.76 \pm 1.39$ | $37.40 \pm 1.27$ | **61.37** $\pm 0.53$ | **60.08** $\pm 1.28$ |
| E.coli | $93.71 \pm 0.39$ | $81.31 \pm 0.61$ | $95.36 \pm 0.35$ | $95.95 \pm 0.35$ | $91.82 \pm 0.58$ | $93.50 \pm 0.44$ | $95.57 \pm 0.44$ | $62.49 \pm 1.43$ | **96.01** $\pm 0.45$ | $95.62 \pm 0.51$ |
| PB | $92.04 \pm 0.35$ | $87.41 \pm 0.39$ | $92.36 \pm 0.34$ | $92.46 \pm 0.37$ | $90.14 \pm 0.45$ | $92.92 \pm 0.35$ | **93.54** $\pm 0.41$ | $77.08 \pm 0.80$ | **92.94** $\pm 0.22$ | $92.48 \pm 0.35$ |
| Yeast | $89.37 \pm 0.61$ | $89.32 \pm 0.60$ | $89.43 \pm 0.62$ | $89.45 \pm 0.62$ | $82.20 \pm 1.02$ | $92.24 \pm 0.61$ | $92.76 \pm 0.55$ | $91.49 \pm 0.57$ | **94.08** $\pm 0.32$ | **95.04** $\pm 0.32$ |
| C.ele | $85.13 \pm 1.61$ | $80.19 \pm 1.64$ | $86.95 \pm 1.40$ | $87.49 \pm 1.41$ | $74.79 \pm 2.04$ | $86.34 \pm 1.89$ | **90.32** $\pm 1.49$ | $77.07 \pm 2.00$ | $88.74 \pm 1.62$ | **89.20** $\pm 1.62$ |

# A  Comparison to heuristic methods for link prediction

To compare our proposed methods with additional popular heuristics methods (Jaccard (Jac.), preferential attachment (PA), Katz, PageRank (PR), and SimRank (SR)) beyond overlapped neighbors-based heuristics, we further conduct extensive experiments on seven traditional link prediction datasets, USAir [1], Power [2], Router [3], E.coli [4], PB [5], Yeast [6], and C.ele [2], used by SEAL [7].

**Datasets** USAir [1] is a network of US Air lines with 332 nodes and 2,126 edges. PB [5] is a network of US political blogs with 1,222 nodes and 16,714 edges. Yeast [6] is a protein-protein interaction network in yeast with 2,375 nodes and 11,693 edges. C.ele [2] is a neural network of C. elegans with 297 nodes and 2,148 edges. Power [2] is an electrical grid of western US with 4,941 nodes and 6,594 edges. Router [3] is a router-level Internet with 5,022 nodes and 6,258 edges. E.coli [4] is a pairwise reaction network of metabolites in E. coli with 1,805 nodes and 14,660 edges.

**Evaluation results** Table 1 shows link prediction results of the heuristic methods and Neo-GNNs on tradiational link prediction datasets. Neo-GNN consistently shows better performance than overlapped-based heuristic methods. Also, we can see that Neo-GNN overall shows better performance than other heuristic methods, except that Pagerank performed better than Neo-GNN in two datasets. Interestingly, though overlap-based heuristic methods perform worse in Power dataset, our Neo-GNN show the best performance compared to all heuristic methods. This result shows that Neo-GNN is not limited to the limitations of existing neighborhood-overlap based heuristics. Although the performance was overall better than other heuristic methods, there is still a limit to generalize higher-order heuristic methods such as Pagerank and Katz index that directly use distance information. The direction of generalizing these heuristic methods will be a good future work.

# B  Analysis on the correlation between the adjacency matrix and the multi-hop adjacency matrix

We observe that the improvement by our Neo-GNNs is more significant especially when the correlation between the original adjacency matrix and the multi-hop adjacency matrix is high. Let $A \in \mathbb{R}^{N \times N}$ denote an adjacency matrix. Then multi-hop adjacency matrix $A' \in \mathbb{R}^{N \times N}$ is defined as

$$A'_{ij} = \begin{cases} 1 & \text{if } \left(\sum_{k=2}^{K} A_{ij}^k\right) > 0 \\ 0 & \text{otherwise.} \end{cases}$$

$A'_{ij} = 1$ means that there is at least one path connecting two nodes in K hops, which also means that overlapped neighbors between two nodes exist within K/2 hops. Then the correlation between $A$ and $A'$ can be computed by $corr(A, A') = \rho(vec(A), vec(A'))$, where $\rho(\cdot, \cdot)$ denotes Pearson Correlation Coefficient between two vectors and $vec(\cdot)$ denotes the vectorization of a matrix. The high correlation between $A$ and $A'$ implies that if a link (or edge) exists between two nodes, then the two nodes have overlapped neighbors and vice versa. In contrast, the closer the correlation is to 0, the less useful it is to predict the actual link with overlapped neighbors. For example, the correlation $\rho$ between $A$ and $A'$ in the training sets of DDI and Collab are about 0.0038 and 0.8727 respectively. This indicates that our Neo-GNN is more effective on Collab rather than DDI. Our experiments in Table 3 are consistent with our analysis.

## C   Broader Impact

Our work focuses on how graph neural networks effectively utilize graph structures in link prediction. Unlike the conventional GNNs that heavily rely on smoothed node features for link prediction, our Neo-GNNs predict links by learning and utilizing structural information using only the graph structures. This implies that our Neo-GNNs can be of great help in various graph-structured data that have only graph structures without node features. Especially, in recommendation systems, which is the most representative application field in link prediction, it is a sensitive issue to exploit users' personal information that can be node features. Even in this case, our Neo-GNNs can safely make recommendations by utilizing useful structural features only through the graph structures. Also, in the biology domain, various relationships between drugs or proteins already exist, but the features for each drug or protein are absent since expensive cost of feature engineering by domain experts are required. Neo-GNNs can be used to discover meaningful relationships without features in the biology domain. Besides, Structural Feature Generator in our Neo-GNNs generates structural features of each node using an adjacency matrix whereas GNNs use input node features. We believe that the Structural Feature Generator will benefit a variety of graph-related tasks beyond link prediction as a model-agnostic module. However, Neo-GNNs need to be used carefully for link prediction in social networks where privacy and anonymity is important.

## D   License of the assets

Our source code is implemented based on PyTorch which was released under Berkeley Software Distribution (BSD) License. We implement various GNN-based baselines using PyTorch Geometric, a graph-specified deep learning framework licensed under MIT. Additionally, we implement SEAL from official GitHub repository [1] under MIT License. Both BSD license and MIT license can be used or redistributed under stipulated conditions. Moreover, we conduct experiments on four benchmark datasets from Open Graph Benchmark. Open Graph Benchmark is released under MIT License. We visualize significant results by using Matplotlib and its license is based on Python Software Foundation (PSF) license which is a permissive free software license.