# OpenReview forum: "Neo-GNNs: Neighborhood Overlap-aware Graph Neural Networks for Link Prediction"
_NeurIPS.cc/2021/Conference — NeurIPS 2021 Poster_

### Official Review · Reviewer_cNTF · 2021-07-10

**Rating:** 6
**Confidence:** 5

**Summary:**

This paper proposes a neighborhood-overlap-aware GNN for link prediction. Traditional GNNs cannot detect the overlap of two nodes when combining the two embeddings as a link representation, which is crucial for link prediction since many successful heuristics such as common neighbor relies on such overlap measure. SEAL leveraging a distance-based node labeling solves this problem (detecting overlap and correlation between two nodes), yet requires extracting a subgraph for each target link thus is computationally more expensive than traditional GNNs. This paper tackles the problem by explicitly combining a learnable overlap measurement function (to simulate overlap-based heuristics) and a traditional GNN (to learn node features), and achieves highly competitive performance with SEAL and outperforms heuristic methods.

**Limitations And Societal Impact:**

One potential concern is that overlap-based heuristics are only one specific type of successful heuristics for link prediction. There are other types of heuristics such as PPR, preferential attachment, etc, which are not based on overlaps. The proposed method generally cannot learn such heuristics in my opinion. I suggest doing another set of experiments on the traditional link prediction datasets used in the SEAL paper to demonstrate the limitation of the proposed method for learning general heuristics. It may be the case that the OGB datasets happen to be very suitable for overlap-based heuristics. I suggest pointing out when the proposed method works well and when it does not.

I give a weak accept conditioned that the authors can address the "Clarity" in the main review and discuss the limitation of the proposed method. I will increase my score accordingly if the authors well address these concerns.

**Main Review:**

Originality: The proposed method can learn node overlapping, which is not feasible in traditional GNNs. It also avoids SEAL's subgraph extraction step, thus is more efficient. The combination of a learnable neighborhood overlap function and a GNN is novel, and works well on the link prediction problem over several OGB datasets.

Quality: The paper is technically sound. The techniques are simple and effective.

Clarity: The paper is generally well written. However, there are several mistakes and inaccuracies in the statements. 1) "SEAL requires inefficient computational cost to manually measure the relative distance for each target node pair and still shows limited performance compared to heuristic methods". This is not true. SEAL outperforms almost all heuristics in its original paper, and the bottleneck of SEAL's computation is applying a GNN independently to each extracted subgraph. The distance calculation is negligible. 2) "However, there is a limitation of the expressive power that is upper-bounded by the 1-Weisfeiler-Lehman (1-WL) graph isomorphism test. That is, some subgraphs are indistinguishable if they are symmetric or isomorphic" This is wrong. We **want** isomorphic subgraphs to be indistinguishable by GNNs. However, some non-isomorphic graphs indistinguishable by 1-WL are also indistinguishable by GNNs lead to the expressivity bottleneck. 3) "Random walk-based embedding methods such as Deepwalk [37], LINE [38], and node2vec [39] ..." LINE is not based on random walk, but first and second-order neighborhood. 4) "Note that conventional GNNs cannot compute this score due to two reasons: the normalized adjacency matrix and the low dimension of hidden representations." This is not true. Conventional GNNs cannot compute the overlap because they independently learn two nodes' neighborhood without modeling their correlations. See [1]. You may perform an experiment replacing the normalized adjacency matrix with unnormalized one in conventional GNNs and increase the hidden representations to verify whether the statement is correct. 5) Eq (8) is not an "aggregation" of neighbor features (although it looks like). It is simply lifting the node features to the corresponding edges. I would not call this an aggregation.

Significance: This paper illustrates the importance of detecting overlap in link prediction, which is significant.

[1] Zhang, M., Li, P., Xia, Y., Wang, K., & Jin, L. (2020). Revisiting graph neural networks for link prediction. arXiv preprint arXiv:2010.16103.

**Time Spent Reviewing:**

3

---

> ### Author Response · Authors · 2021-08-10
> **Response to Reviewer 3**
>
> Thank you very much for taking the time to review our submission and the comments! We are happy to respond to your comments as below:
>
> >Q1. "SEAL requires inefficient computational cost to manually measure the relative distance for each target node pair and still shows limited performance compared to heuristic methods". This is not true. SEAL outperforms almost all heuristics in its original paper, and the bottleneck of SEAL's computation is applying a GNN independently to each extracted subgraph. The distance calculation is negligible.
>
> The inefficiency of SEAL we addressed was that the relative distance had to be calculated **for each target pair**. But as you pointed out, we agree that applying GNN to each extracted subgraph causes the biggest bottleneck in SEAL.
> Also, considering the experimental results in SEAL’s original paper, we agree that SEAL performs better than most of the heuristic methods in traditional link prediction datasets.
>
> Thanks for your clarification and we’ll replace the existing part of the limitation of SEAL with an inefficiency of applying GNN to each extracted subgraph in the final version.
>
>
> > Q2. "However, there is a limitation of the expressive power that is upper-bounded by the 1-Weisfeiler-Lehman (1-WL) graph isomorphism test. That is, some subgraphs are indistinguishable if they are symmetric or isomorphic" This is wrong. We want isomorphic subgraphs to be indistinguishable by GNNs. However, some non-isomorphic graphs indistinguishable by 1-WL are also indistinguishable by GNNs lead to the expressivity bottleneck.
>
> You are right. The sentence is misleading. The limitation of conventional GNNs has been studied with some non-isomorphic graphs indistinguishable by 1-WL. We will clarify this in the final version.
>
>
> > Q3. "Random walk-based embedding methods such as Deepwalk [37], LINE [38], and node2vec [39] ..." LINE is not based on random walk, but first and second-order neighborhood.
>
> Thanks for pointing out an error in the related works. We will clarify this in the final version.
>
>
> >Q4. "Note that conventional GNNs cannot compute this score due to two reasons: the normalized adjacency matrix and the low dimension of hidden representations." This is not true. Conventional GNNs cannot compute the overlap because they independently learn two nodes' neighborhood without modeling their correlations. See [1]. You may perform an experiment replacing the normalized adjacency matrix with unnormalized one in conventional GNNs and increase the hidden representations to verify whether the statement is correct.
>
> The statement is misleading. The word "the low dimension of hidden representations" was intended to indicate conventional GNNs do not use a diagonal matrix. We will correct this in the revised version.
>
> Unlike conventional GNNs, our model uses a diagonal matrix to separate information between nodes and utilizes an unnormalized adjacency matrix to hinder information being normalized. Hence, Neo-GNN is able to estimate the overlapped neighborhood without modeling correlations.
> We also agree that conventional GNNs are not able to measure neighborhood overlap because GNNs independently learn nodes' neighborhoods without modeling their correlations .
> By referring to the paper [1], we will clarify this point in the final version.
>
>
> > Q5. Eq (8) is not an "aggregation" of neighbor features (although it looks like). It is simply lifting the node features to the corresponding edges. I would not call this an aggregation.
>
> This is a good point. Since $X^{struct}$ is a diagonal matrix, it is correct that Eq (8) lifts the node features to the corresponding edges. However, since the dimension of each node feature vector is N (i.e, $x^{struct}_{i} \in \mathbb{R}^{N \times 1}$), Eq (8) can also be seen as an aggregation of neighbor features.
> It becomes clear If Eq (8) is expressed in vector-level form as below:
>
> $z_i = \sum_{j \in  \mathcal{N}_i}{x^{struct}_j}$
>
> Also, we explained the Eq (8) as an aggregation since it would be more familiar to our readers. But It is also correct and interesting to be interpreted in terms of lifting the node features to the corresponding edge, so we will add a discussion about how Eq (8) can be interpreted from two perspectives.
>
>
> > Q6. One potential concern is that overlap-based heuristics are only one specific type of successful heuristics for link prediction. There are other types of heuristics such as PPR, preferential attachment, etc, which are not based on overlaps. The proposed method generally cannot learn such heuristics in my opinion. I suggest doing another set of experiments on the traditional link prediction datasets used in the SEAL paper to demonstrate the limitation of the proposed method for learning general heuristics. It may be the case that the OGB datasets happen to be very suitable for overlap-based heuristics. I suggest pointing out when the proposed method works well and when it does not.
>
> Thank you for the helpful suggestion. We agree that our proposed method focuses on learning and generalizing overlap-based heuristics.
> To identify the limitations of our method for learning general heuristics, we have measured performance of our Neo-GNN on 8 traditional link prediction datasets used in the SEAL paper.
> The performance of our Neo-GNNs is the mean and the standard deviation of performances from 10 independent runs. The performance of heuristic methods is the same as presented in the paper. The results are as follows:
>
>
> |        |       ||       |       |       |       |       |       |       |                  |
> |--------|:----------:|:-------:|:-----------:|:---------:|:-------:|:-------:|:-------:|:-------:|:----------:|:----------------:|
> |        | CN       | Jac.  | AA        | RA      | PA    | Katz  | PR    | SR    | Neo-GNN| Neo-GNN |
> |        |        |   |       |      |    |   |     |     | (w/o GCN) | (w/GCN) |
> | USAir  | 93.80±1.22    | 89.79±1.61 | 95.06±1.03  | 95.77±0.92  | 88.84±1.45 | 92.88±1.42 | 94.67±1.08 | 78.89±2.31 | **96.10**±0.79            | 95.56±0.57          |
> | Power  | 58.80±0.88    | 58.79±0.88 | 58.79±0.88     | 58.79±0.88   | 44.33±1.02 | 65.39±1.59 | 66.00±1.59 | 76.15±1.06 | **76.17**±0.87            | 72.42±1.34          |
> | Router | 56.43±0.52    | 56.40±0.52 | 56.43±0.51     | 56.43±0.51   | 47.58±1.47 | 38.62±1.35 | 38.76±1.39 | 37.40±1.27 | **61.37**±0.53        | 60.08±1.28     |
> | E.coli | 93.71±0.39    | 81.31±0.61 | 95.36±0.35     | 95.95±0.35   | 91.82±0.58 | 93.50±0.44 | 95.57±0.44 | 62.49±1.43 | **96.01**±0.45       | 95.62±0.51          |
> | PB     | 92.04±0.35    | 87.41±0.39 | 92.36±0.34     | 92.46±0.37   | 90.14±0.45 | 92.92±0.35 | **93.54**±0.41 | 77.08±0.80 | 92.94±0.22            | 92.48±0.35          |
> | Yeast  | 89.37±0.61    | 89.32±0.60 | 89.43±0.62     | 89.45±0.62   | 82.20±1.02 | 92.24±0.61 | 92.76±0.55 | 91.49±0.57 | 94.08±0.32            | **95.04**±0.32          |
> | C.ele  | 85.13±1.61    | 80.19±1.64 | 86.95±1.40     | 87.49±1.41   | 74.79±2.04 | 86.34±1.89 | **90.32**±1.49 | 77.07±2.00 | 88.74±1.62            | 89.20±1.62          |
>
> Neo-GNN consistently shows better performance than overlapped-based heuristic methods. Also, we can see that Neo-GNN overall shows better performance than other heuristic methods, except that Pagerank performed better than Neo-GNN in two datasets.
>
> Interestingly, though overlap-based heuristic methods perform worse in Power dataset, our Neo-GNN show the best performance compared to all heuristic methods. This result shows that Neo-GNN is not limited to the limitations of existing neighborhood-overlap based heuristics.
> Although the performance was overall better than other heuristic methods, there is still a limit to generalize higher-order heuristic methods such as Pagerank and Katz index that directly use distance information. The direction of generalizing these heuristic methods will be a good future work.

---

> > ### Comment · Reviewer_cNTF · 2021-08-17
> > **Comments to author response**
> >
> > I thank the authors for the added experiments. Please add these experiment results and point out the limitations of the proposed method in the revised paper. I encourage the authors to explore fast learning algorithms for non-overlap-based heuristics too, and expect to see a larger impact of the paper bringing to the link prediction community.

---

> > > ### Author Response · Authors · 2021-08-27
> > > **Response to Reviewer 3**
> > >
> > > Thank you for your response on our rebuttal! We are happy with the fact that our response addressed "the Clarity in the main review" and discussed "the limitation of the proposed method". We hope that this leads to the stronger support. We are grateful for your help to improve our paper.

---

### Official Review · Reviewer_yBWB · 2021-07-14

**Rating:** 7
**Confidence:** 4

**Summary:**

The paper proposes to use simple functions learned from the adjacency matrix of a graph to boost the performance of GNNs. This overcomes the problem of using traditional heuristics which are effective but must be computed manually. The proposed approach is simple and the experiments over the quite challenging datasets of the OGB show that it is rather effective. I have some concerns regarding the latter which is why my current score is only slightly below the threshold, but maybe the authors can resolve those.

**Ethical Concerns:**

None.

**Limitations And Societal Impact:**

Yes.

**Main Review:**

(+)

- The proposed approach is simple but effective which is quite nice. It is so simple that I wonder if there is existing work which the authors and I are missing, but maybe the other reviewers can confirm that this is not the case.

- Generally, the paper is well-written and I have few to criticise. The approach is described in detail, related work is covered, and the experiments are overall thorough. In particular, the latter show that their learned functions correlate with well-known traditional heuristics.

(-)

- To me, it is not clear why the authors use a GNN for learning from the features. In particular, they argument a lot that GNNs fail to learn the structure (which is a strong claim btw), so why use a GNN at all and not a simple MLP?

- The SAGE number reported in the table is worse than the one reported in the OGB paper. Did you try with the parameters they use there? There is further GraphSAGE+anchor distance on the leaderboard, whose ogbl-ddi score (0.8239) is considerably better than the one for NeoGNN. And it is quite related given that it also considers a structural feature.

- 260: "Neo-GNNs (w/o GCN) still show the state-of-the-art performances compared to baselines except OGB-DDI. Note that Neo-GNNs (w/o GCN) only use graph structures and outperform other GNNs whereas other GNNs use both input features and graph structures.." -- To the best of my knowledge, it is the DDI dataset that does not contain node features, but uses randomly initialized ones. In this regard, I wonder about the conclusion of the authors. To me it seems that the other GNNs obviously manage to use the structure to make something out of the random initial features.

- I do not think it is fair to fix the numbers of GNN layers to 3. Especially over these OGB datasets, my personal experience shows that they work often better with several more layers. And at least JK-Net should be able to achieve good scores (while others might suffer from over-smoothing). Did you get similar scores with more layers?

-------------------------------
Questions & Smaller Comments
-------------------------------

- Figure 2: Did you check if the trends are similar on the other datasets?
- Which L is used in Figure 2b?
- 107: s does not necessarily have to be a similarity function
- 166: Do you really count the number of neighbours here?
- 169ff: Is this just additional information or important in some regard. To me, the remark is somehow confusing.
- 199: You mean multiple edge types? (edges of course)
- I think the LINK model from Zheleva and Getoor ("To join or not to join: The illusion of privacy in social networks with mixed public and private user profiles", 2009) would be worth mentioning as related work.

-------------------------------
edit: I adapted the score based on the responses in the rebuttal

**Time Spent Reviewing:**

2

---

> ### Author Response · Authors · 2021-08-10
> **Response to Reviewer 2**
>
> Thank you very much for taking the time to review our submission and the comments! We are happy to respond to your comments as below:
>
> > Q1.  To me, it is not clear why the authors use a GNN for learning from the features. In particular, they argument a lot that GNNs fail to learn the structure (which is a strong claim btw), so why use a GNN at all and not a simple MLP?
>
> This is a great question. As we stated in our paper, conventional GNNs heavily rely on **smoothed node features** rather than graph structure, which fail to learn key structural information for link prediction. However, this also means that GNNs are still effective for learning feature-based representations when an input graph is homophily. Since the datasets we used did not provide node labels, we can’t compute the homophily ratio, but our experimental results show that GCN consistently performs better than MLP. Therefore, we use a GNN for learning from the features.
>
> |     |    OGB-PPA   |  OGB-COLLAB  |    OGB-DDI   | OGB-CITATION2 |
> |-----|:------------:|:------------:|:------------:|:-------------:|
> | MLP | 0.47 ± 0.05  | 19.98 ± 0.96 |      N/A     | 28.99 ± 0.16  |
> | GCN | 16.98 ± 1.33 | 47.01 ± 0.79 | 44.60 ± 8.87 | 84.79 ± 0.24  |
>
> > Q2. The SAGE number reported in the table is worse than the one reported in the OGB paper. Did you try with the parameters they use there?
>
> Yes, we used official codes for each baseline model from the OGB leaderboard and also tried the same parameters they used. However, in DDI dataset, the performance changed even with the same code and we couldn’t get the same performance of GraphSAGE reported in the leaderboard. The performance varies greatly depending on each seed as random initialized vectors are used as node features. To solve this issue, we evaluated performances with a seed fixed to 0~9 as OGB recommends, so even if there are differences with the performance in the leaderboard, models in our experiments were compared fairly in the same seed set.
>
> > Q3. There is further GraphSAGE+anchor distance on the leaderboard, whose ogbl-ddi score (0.8239) is considerably better than the one for NeoGNN. And it is quite related given that it also considers a structural feature.
>
> First, we'd like to emphasize that GraphSAGE+anchor distance model was published on the leaderboard 6 days before the NeurIPS submission deadline, June 26th, and It was hard to consider this model while working on the paper.
>
> Although the model obtained excellent performance, the model requires an expensive cost to manually compute the distance of all links, and this structural information is just used in the prediction layer independent of GraphSAGE. GraphSAGE+anchor is a model that predicts a link using the distance information between two nodes and two node representations from the GraphSAGE. This model manually computes distances of all the given links as follows:
> 1. Randomly pick K anchor nodes, $a_1, a_2, ..., a_{K}$ and compute distances of all the given links as $\text{dist}_{u,v} = \frac{1}{K} \sum^K_i{\text{dist}(u, a_i)+\text{dist}(a_i, v)} $
>
> Since, in DDI dataset, they use 1000 anchor nodes among 4267 nodes in the graph, the time complexity of computing all distances is $O(K|\mathcal{E}|^2|V|))$~$O(|\mathcal{E}|^3|V|))$ since $K=\frac{1}{4}|\mathcal{E}|$ and the time complexity of the single-source shortest path is $O(|V||E|)$. It requires an expensive cost and we have the concern about the scalability of this model.  Also, as R1 pointed out, this is a trivial extension (combination) of the existing methods whereas our approach integrated the structure information in GNNs.
>
> > Q4. 260: "Neo-GNNs (w/o GCN) still show the state-of-the-art performances compared to baselines except OGB-DDI. Note that Neo-GNNs (w/o GCN) only use graph structures and outperform other GNNs whereas other GNNs use both input features and graph structures.." -- To the best of my knowledge, it is the DDI dataset that does not contain node features, but uses randomly initialized ones. In this regard, I wonder about the conclusion of the authors. To me it seems that the other GNNs obviously manage to use the structure to make something out of the random initial features.
>
> We’d like to restate the first sentence “Neo-GNNs (w/o GCN) still show the state-of-the-art performances compared to baselines **except OGB-DDI**”. Neo-GNNs (w/o GCN) doesn’t outperform GNN baseline in the DDI dataset, so the statement "Neo-GNNs (w/o GCN) outperforms other GNNs without node features" is related to other datasets except for the DDI dataset. However, the sentence is not clear, causing misunderstanding, so we will clarify the mentioned sentences clearly in the final version.
>
> > Q5. I do not think it is fair to fix the numbers of GNN layers to 3. Especially over these OGB datasets, my personal experience shows that they work often better with several more layers. And at least JK-Net should be able to achieve good scores (while others might suffer from over-smoothing). Did you get similar scores with more layers?
>
> This is a good point. In our experience, the overall performance of GNN baselines did not change significantly after layer 3 or rather decreased. To address your concerns about fixing the numbers of GNN layers, we added the additional experimental results of JK-Net according to the number of layers as follows.
>
> |        |    Layer 3   |    Layer 4   |    Layer 5   |   Layer 6   |    Layer 7   |
> |--------|:-----------:|:-----------:|:-----------:|:----------:|:-----------:|
> |   DDI  | 57.98±6.88 | 49.76±5.88 | 34.83±6.96 |     -       |      -       |
> | Collab | 46.34±0.65 | 47.55±0.67 |  48.23±0.9 | 48.41±0.5 | 48.84±0.83 |
> |   PPA  | 11.25±1.32 | 11.06±5.37 | 11.40±2.04  |     -       |      -       |
>
> As can be seen from the above results, the performance of JK-Net decreases or slightly increases as the number of layers increases in DDI and PPA. Although there is a performance gain in Collab, it is still lower than the performance of our Neo-GNN (57.52 ± 0.37).
> But we agree that the optimal number of layers can be different for each GNN and we will update the performance of JK-Net in Collab based on layer 7.
>
> * Questions & Smaller Comments
>
> >Figure 2: Did you check if the trends are similar on the other datasets?
>
> Yes, we checked. For all datasets, increasing number of layers $L$ was helpful for performance, but the scale of the optimal $\beta$ was different according to the sparsity of each graph dataset. We will add the ablation study of $L$ and $\beta$ in other datasets in the final version.
>
>
> > Which L is used in Figure 2b?
>
> We used L as two.
>
>
> >107: s does not necessarily have to be a similarity function
>
> Thanks for your suggestion. We’ll clarify the definition of s in the final version.
>
>
> > 166: Do you really count the number of neighbours here?
>
> Sorry for the confusion. What we meant to say was to “consider” the number of neighbors. We will clarify this in the revised paper.
>
>
> >169ff: Is this just additional information or important in some regard. To me, the remark is somehow confusing.
>
> The remark on this part is actually to show mathematically how our model can obtain information about overlapped neighbors between two nodes.
>
>
> >199: You mean multiple edge types? (edges of course)
>
> No. Multiple edges here refers to overlapping edges between two nodes. In other words, it meant that there were overlapping edges, which meant that A_ij could be a value greater than 1, not just 0 or 1.
>
>
> >I think the LINK model from Zheleva and Getoor ("To join or not to join: The illusion of privacy in social networks with mixed public and private user profiles", 2009) would be worth mentioning as related work.
>
> Thanks for suggesting the reference. We will add the discussion about the LINK in the final version.

---

> > ### Comment · Reviewer_yBWB · 2021-08-20
> > **Response**
> >
> > I acknowledge that I have read the other reviews and responses and thank the authors for the detailed explanations. All my questions were addressed.
> >
> > While I agree that the paper would have to clarify several inaccuracies in the writing in the final version, the approach is nice. I do not agree with Reviewer 9RGy in that trivial solutions are basically too trivial for NeurIPS. In contrast, if simple techniques provide the solutions to what we need, it is great to point this out by providing sufficient evidence instead of suggesting ever more complex models which we do not understand.

---

> > > ### Comment · Reviewer_9RGy · 2021-08-20
> > > **"trivial"**
> > >
> > > Thanks for reading reviews. I want to clarify my point about "trivial".
> > >
> > > First of all, I did not say "trivial solutions are basically too trivial for NeurIPS." This is originally created by Reviewer yBWB. Actually I like simple techniques. I agree that effective simple techniques are desired.
> > >
> > > Second, let me quote the only place I said "trivial" : "Creating two models and connecting at the last layer is probably the most trivial idea." It looks for the motivation of joint learning and its design. Like in many other places of my review, I don't feel the motivation of joint learning is clear. Simply say, why the two (in fact, three) objective terms are complementary in the optimization; why not use one to replace the other.
> > >
> > > Third, I feel the design of joint learning is not yet clear. Like in my further comments (see the second point), I am looking for the concrete shared parameters in the joint learning process. I hope the authors could address my confusion.
> > >
> > > As long as the authors could clarify the motivation of joint learning ("why complementary") and the design of joint learning ("how the knowledge was shared/transferred"), I would be happy to update my review. Again, I've never favored complex solutions over simple ones. I favor clear, well motivated, and solid solutions.

---

> > > ### Author Response · Authors · 2021-08-27
> > > **Response to Reviewer 2**
> > >
> > > Thank you for your response on our rebuttal! We are happy with the fact that our response addressed all your concerns and really appreciate your comments on this paper.

---

### Official Review · Reviewer_9RGy · 2021-07-14

**Rating:** 5
**Confidence:** 4

**Summary:**

This work added a model into GNN to improve link prediction. The connection is a weighted combination of the model's output and GNN's output in the scoring function (scoring a predicted link).  The paper claims that this model computes the number of overlapped neighbors. Experiments show the combination improves the performance.

**Ethical Concerns:**

I don't have any concerns.

**Limitations And Societal Impact:**

Not included or discussed in the paper.

**Main Review:**

Originality: It is an incremental approach with lots of things that are not clearly explained.

- Eq.(8) aggregates features of neighbors but why it can "count the number of overlapped neighbors"?
- What is "overlapped neighbors"?  It seems to be a pair-wise concept that measures the relationship between two nodes (overlapped neighbors of the two nodes). However, I am not seeing any pair-wise measurement in the equations.
- Why choose the number of overlapped neighbors, when there has been multiple kinds of heuristics-based measurements (as discussed in Section 3.2)?
- There should be more essential methods to make a GNN model be aware of overlapped neighbors/subgraph.  Creating two models and connecting at the last layer is probably the most trivial idea. I suggest the authors to incorporate the overlapping subgraph into the convolutional operation in Eq.(10).
- Even though the models could be connected and jointly trained, it is not clear whether the model parameters are shared. Is \Theta in Eq.(9) used in Eq.(10)?

Quality: I have serious concern about the motivation of the work and thus feel the quality of the proposed method and experimental results is questionable.

Clarity: The incremental idea is not hard to follow.

Significance: Very little impact. There was not theoretical analysis that proved that the structural information about overlapping neighbors would be hardly learned by the existing GNN.

*** The authors have clarified all of the points with either new results or explanations in the multi-rounds discussion. No matter the work is accepted or further reviewed, the authors are highly encouraged to add the new results and explanations into the paper. ***


**Time Spent Reviewing:**

1 hour

---

> ### Author Response · Authors · 2021-08-10
> **Response to Reviewer 1**
>
> Thank you very much for taking the time to review our submission and the comments! We are happy to respond to your comments as below:
>
> > Q1. There should be more essential methods to make a GNN model be aware of overlapped neighbors/subgraph. Creating two models and connecting at the last layer is probably the most trivial idea. I suggest the authors to incorporate the overlapping subgraph into the convolutional operation in Eq.(10).
>
> Thank you for the suggestion. Indeed, our proposed model is exactly what you suggested. Our main contribution is proposing a new GNN that is capable of estimating overlapped neighborhoods. We show how we incorporate the overlapping subgraph estimation into graph convolution by mapping each component of Neo-GNN to the basic components of GNN.
>
> 1. Input feature generation from the adjacency matrix by Structural Feature Generator in Eq. (6), i.e., $\mathbf{x}^{\text{struct}}=\mathcal{F}_{\theta}(A)$
> 2.  Constructing a diagonal matrix from the scalar value can be viewed as a node-specific linear transformation Eq. (7), $X_{\text{struct}} = \text{diag}(\mathbf{x}^{\text{struct}}) \iff \mathbf{x}^{\text{struct}} 1^T \odot I$. It is similar to XW in GCN.
>
> 3.  Aggregation of features of neighborhoods and node itself via a multi-hop way (Eq. (8), Eq. (9)).
>
>     $Z = g_{\Phi} \left (\sum_{l=1}^{L}{\beta^{l-1} A^{l}} X^{struct} \right )$. Recently, similar multi-hop aggregation has been used in MixHop, and GRAND to mixing features of neighbors at various distances.
>
>
> Since we constructed the diagonal matrix from node features, our Neo-GNN can estimate the overlapped neighbors by the simple inner product of two nodes' representations $z_i^T z_j$ as Eq. (11)
>
>
> > Q2. Eq.(8) aggregates features of neighbors but why it can "count the number of overlapped neighbors"?
>
> As we mentioned above, the **diagonal matrix** in Eq. (8) allows us to estimate overlapped neighbors by the simple inner product of two node representations $z_i^T z_j$ after aggregation.
>
> Further, to count/consider the **number** of overlapped neighbors, we use an **unnormalized** adjacency matrix in Eq. (8) instead of the normalized adjacency matrix used in the conventional GNNs.
> Our model can calculate the number of overlapped neighbors, but what we meant in "count the number of overlapped neighbors" was that our model could consider number of overlapped neighbors with unnormalized adjacency matrix.
>
> The word 'count' can be confusing enough, so We'll fix it with another word in the revised paper.
>
> > Q3. What is "overlapped neighbors"? It seems to be a pair-wise concept that measures the relationship between two nodes (overlapped neighbors of the two nodes). However, I am not seeing any pair-wise measurement in the equations.
>
> "Overlapped neighbors" denotes the common neighbors of given two nodes. This information has been proven crucial to link prediction. While previous GNN-based models could not explicitly estimate the overlapped neighbors without hand-crafted features, e.g., distance encoding, our Neo-GNNs can measure the overlapped neighbors through node features automatically extracted from the adjacency matrix and their inner products. Line 171 and Eq (11) show how the pair-wise measurement is computed.
>
> > Q4. Why choose the number of overlapped neighbors, when there has been multiple kinds of heuristics-based measurements (as discussed in Section 3.2)?
>
> Your question is exactly the motivation of our proposed method. We propose a model that can **generalize** heuristic methods in Sec 3.2 and automatically **learn** the optimal heuristic methods for each dataset. Specifically, as we described in line (154-156), Structural feature generator $F_{\theta}$ in Eq. (6) can generate structural features for each heuristic method. For example, If $f_{θ_{node}}$ is a reciprocal of the logarithm function, i.e., $f(x)=\frac{1}{log(x)}$ , and $f_{\theta_{edge}}$ is an identity function, i.e., $f(x)=x$, then Structural feature generator $F_{\theta}$ can generate the exactly same structural feature as the features used in Adamic-Adar method.
> Also, we experimentally showed that our Neo-GNN can learn each heuristic methods in Sec 4.4 and Fig. 3.3 .
>
>
> > Q5. Even though the models could be connected and jointly trained, it is not clear whether the model parameters are shared. Is $\theta$ in Eq.(9) used in Eq.(10)?
>
> No, parameters $\theta$ and $\Phi$ in our Neo-GNN are not shared with the conventional GNN in Eq. (10). $\theta$ is used for structural feature generation from the adjacency matrix by Structural Feature Generator in Eq. (6) whereas the parameter $W$ in GNN is used for linear transformation of given input features.

---

> > ### Comment · Reviewer_9RGy · 2021-08-18
> > **On Q3/Q4 and Q5**
> >
> > Thanks for the authors' clarification on Q1 and Q2. They are helpful for understanding the paper. I have questions upon Q3/Q4 and Q5.
> >
> > Q3/Q4: [overlap neighborhood / common neighbors]
> > It is common to see long tails in degree distributions on real graphs. The matrix of number of common neighbors can be even sparser. The value of the common neighbor counts would be an extreme long-tail distribution. So if the inner product of a pair of node representations (in Eq.(11)), after nonlinear transformation, is estimating the count of common neighbors, I would suspect the informativeness of the representations. To the worse case, they could have many zeros. Empirical analysis, and to the best, theoretical analysis are desired. Is the improvement brought by neighborhood-overlap aware learning more significant on denser graph or sparser graph, active nodes (node pairs) or inactive nodes? Suppose a practitioner has a new graph and is able to calculate many kinds of statistics of the graph. How could he/she know whether the neighborhood-overlap aware learning would be effective?
> >
> > Q5: Eq.(10) creates H/h. Eq. (9) creates Z/z. If the model parameters were not shared, the (three) terms in the "joint training loss function" (Eq.(12)) would be isolated and trained only upon themselves. Can the authors clarify how the joint training could transfer knowledge across the objectives? Any critical model parameters are shared and jointly learned?

---

> > > ### Author Response · Authors · 2021-08-27
> > > **Response to Reviewer 1**
> > >
> > > Thank you for your response on our rebuttal! Please see our response in detail:
> > >
> > > > Q3/Q4: [overlap neighborhood / common neighbors] It is common to see long tails in degree distributions on real graphs. The matrix of number of common neighbors can be even sparser. The value of the common neighbor counts would be an extreme long-tail distribution. So if the inner product of a pair of node representations (in Eq.(11)), after nonlinear transformation, is estimating the count of common neighbors, I would suspect the informativeness of the representations. To the worse case, they could have many zeros. Empirical analysis, and to the best, theoretical analysis are desired. Is the improvement brought by neighborhood-overlap aware learning more significant on denser graph or sparser graph, active nodes (node pairs) or inactive nodes? Suppose a practitioner has a new graph and is able to calculate many kinds of statistics of the graph. How could he/she know whether the neighborhood-overlap aware learning would be effective?
> > >
> > > This is a great question!
> > >
> > > First, our framework works well on sparse graphs since our Neo-GNNs consider **_multi-hop overlapped neighbors_** (Eq. (9)). As you said, the number of 1-hop common neighbors (or 2-hop paths) between any two nodes is mostly zero if the graph is extremely sparse. For instance, the graph in Collab dataset is relatively sparse and the density is 0.0046%. As a result, the ratio of node pairs with overlapped neighbors is also as low as 0.05%. However, considering up to 2-hop overlapped neighbors (or up to 4-hop paths), the ratio increases by almost 100 times to 4.95%. More importantly, the node pairs with 2-hop overlapped neighbors cover about 86% of unseen links in the test dataset.
> > >
> > > Second, we observe that the improvement by our Neo-GNNs is more significant especially when the correlation between the original adjacency matrix and the multi-hop adjacency matrix is high. Let $A \in \mathbb{R}^{N \times N}$ denote an adjacency matrix. Then multi-hop adjacency matrix $A' \in \mathbb{R}^{N \times N}$ is defined as
> > >
> > > $A'_{ij}= \begin{cases} 1 & \text{if}\ (\sum\_{k=2}^{K}{A^k\_{ij}}) > 0 \\\ 0 & \text{otherwise.} \end{cases}$
> > >
> > > $A_{ij}'=1$ means that there is at least one path connecting two nodes in $K$ hops, which also means that overlapped neighbors between two nodes exist within $\frac{K}{2}$ hops. Then the correlation between A and A' can be computed by $corr(A, A') = \rho(vec(A), vec(A'))$, where $\rho(\cdot, \cdot)$ denotes Pearson Correlation Coefficient between two vectors and $vec(\cdot)$ denotes the vectorization of a matrix.
> > >
> > > The high correlation between A and A' implies that if a link (or edge) exists between two nodes, then the two nodes have overlapped neighbors and vice versa. In contrast, the closer the correlation is to 0, the less useful it is to predict the actual link with overlapped neighbors. For example, the correlation $\rho$ between A and A' in the training sets of DDI and Collab are about 0.0038 and 0.8727 respectively. This indicates that our Neo-GNN is more effective on Collab rather than DDI. Our experiments in Table 3 are consistent with our analysis.
> > >
> > > > Q5: Eq.(10) creates H/h. Eq. (9) creates Z/z. If the model parameters were not shared, the (three) terms in the "joint training loss function" (Eq.(12)) would be isolated and trained only upon themselves. Can the authors clarify how the joint training could transfer knowledge across the objectives? Any critical model parameters are shared and jointly learned?
> > >
> > > Two branches in (9) and (10) do not share any model parameters but the parameters are jointly trained by the first loss term $\lambda\_1\text{BCE}(\hat{y}\_{ij}, y\_{ij})$ in Eq.(12) since $\hat{y}\_{ij}$ is a convex combination of the two branches' outputs, i.e., $\alpha \cdot \sigma(z\_i^Tz\_j) + (1- \alpha) \cdot \sigma(s(h\_i, h\_j))$ in Eq. (11).
> > >
> > > In addition, our model automatically learns how to combine the outputs of two branches by optimizing a learnable parameter $\alpha$. It can be viewed as the relative importance of structural information and feature information in each dataset. As Table 3 and the discussion at Line 265, when the structure information is more useful, e.g., PPA where Neo-GNN (w/o GCN) significantly outperforms GCN, our Neo-GNN (w/ GCN) successfully learns $\alpha\approx 1$ and utilizes the $Z$ in (9) more than $H$ in (10).

---

> > > > ### Comment · Reviewer_9RGy · 2021-08-27
> > > > **$\alpha$**
> > > >
> > > > Many thanks for the clarification to my question on Q3/Q4. That makes a lot of sense. I encourage authors to add it into the paper.
> > > >
> > > > About joint learning -
> > > > The joint learning loss function $\mathcal{L}$ has three objectives - let's name them $L_1$, $L_2$, and $L_3$. So $\mathcal{L} = \lambda_1 L_1 + \lambda_2 L_2 + \lambda_3 L_3$. It seems to me $L_1$ is a generalized form of $L_2$ (when $\alpha=1$) and $L_3$ (when $\alpha=0$). If $\alpha$ is automatically learned, why do we need $L_2$ and $L_3$ (which are two special cases that might not be optimal - $L_1$ owns the optimum of $\alpha$)? If adding such an objective ($L_1$) can improve the performance (though we have $\lambda_2$ and $\lambda_3$ to tune), why not add more and more objectives similar as $L_1$ that use learnable parameters $\alpha_1$, $\alpha_2$ ... to replace $\alpha$? And why should we assess the result $\alpha \approx 1$ to be "successful"? In that case, the (successful) loss function is $(\lambda_1+\lambda_2) L_2 + \lambda_3 L_3$?

---

> > > > > ### Author Response · Authors · 2021-08-31
> > > > > **Response to Reviewer 1**
> > > > >
> > > > > Thank you for actively participating discussion and giving us opportunities to improve our manuscript. We are happy with the fact that our response addressed your concerns and as suggested we will add the analysis about Q3/Q4 in the final version.
> > > > >
> > > > > Regarding the loss function, we provide an additional ablation study justifying our loss function. First, $\alpha$ is not part of the objective function. $\alpha$ is a model parameter to combine predictions from two branches, i.e., $\sigma(z_i^Tz_j)$ and $\sigma(s(h_i, h_j))$. $L_2$ and $L_3$ are necessary for training the two individual branches and $L_1$ is needed for learning how to combine them improving the final prediction $\hat{y}_{ij}$. Our Neo-GNN can be viewed as an ensemble model of the two branches. Our experimental results below show that training without $L_2$ and $L_3$ suffers from noticeable degradation. Especially on OGB-DDI the degradation is about 6.05. This result shows that our loss function with $L_2$ and $L_3$ provides more stable/effective training.
> > > > >
> > > > > |      |  OGB-COLLAB  |    OGB-DDI
> > > > > |-----|:------------:|:------------:|
> > > > > | Neo-GNN ($L_1 + L_2 + L_3$) | 57.52 ± 0.37  | 63.57 ± 3.52 |
> > > > > | Neo-GNN ($L_1$) | 56.89 ± 0.53 | 53.94 ± 6.98 |
> > > > >
> > > > > On PPA in the Table below, $\alpha\approx 1$ is considered as "successful" since our Neo-GNN (w/ GCN) downweighs the erroneous prediction of GCN and generates predictions heavily relying on more trustworthy branch $\sigma(z_i^Tz_j)$ . On the other hand, when both branches are reliable like COLLAB, our Neo-GCN utilizes both branches with $\alpha \approx 0.57$. The learning parameter $\alpha$ may indicate which is more reliable between node feature-based prediction $\sigma(h_i^Th_j)$ and overlapped neighbor-based prediction $\sigma(z_i^Tz_j)$.
> > > > >
> > > > > |  Dataset    |  $\alpha$  |    Neo-GNN  |  Neo-GNN (w/o GCN) | GCN
> > > > > |-----|:------------:|:------------:|:------------:|:------------:|
> > > > > | OGB-PPA | 0.98 ± 0.003  | 49.13 ± 0.60 | 48.63 ± 0.88| 16.98 ± 1.33 |
> > > > > | OGB-COLLAB | 0.57 ± 0.130 | 57.52 ± 0.37 | 55.70 ± 0.24 | 47.01 ± 0.79 |
> > > > > | OGB-DDI | 0.48 ± 0.015 | 63.57 ± 3.52 | 17.38 ± 4.05 | 44.60 ± 8.87 |
> > > > > | OGB-CITATION2 | N/A | OOM | 87.26 ± 0.84 | 84.79 ± 0.24 |

---

> > > > > > ### Comment · Reviewer_9RGy · 2021-09-01
> > > > > > **Feedback**
> > > > > >
> > > > > > Thanks for answering my questions in the last round. There are a few action items to improve the paper besides adding the clarification to my ***earlier questions*** into the paper.
> > > > > > (1) Add sentences to present the "ensembling" trick when introducing the overall objective function. Add the results of using L1 only into Table 2. Mention that both the neighborhood overlap learning and ensembling trick contributed to the improvement from 54.37 to 57.52 (on Collab).
> > > > > > (2) Clearly define what is successful. It is more likely an interpretation of the observation on learned parameters. When looking at $\alpha \approx 0.57$ from experimental results, we could not claim 0.53 or 0.61 would be less successful. It is more like you have a hypothesis on the difference of $\alpha$ between datasets, and the results validate the hypothesis. A hypothesis testing is desired instead of simply judging a value (that has no way to get a "ground truth") to be a success or failure.
> > > > > > I am changing my score from 3 to 5. Through the discussion (which is really useful), we can see quite a lot of to-do items (which will be very helpful for readers). Readers deserve a better paper.

---

### Decision · Program_Chairs · 2021-09-27

**Decision:**

Accept (Poster)

**Comment:**

The paper proposes a novel GNN architecture for link prediction based on neighbourhood overlap.
The authors provided a detailed rebuttal including additional experimental results requested by the reviewers. There has been an extensive follow-up discussion. While doubts remain about clarity of the presentation, the AC believe this can be improved in the final version according to the discussion with the reviewers and their recommendations. We recommend accepting the paper.